# Detection of Multidrug-Resistant *Enterobacterales*—From ESBLs to Carbapenemases

**DOI:** 10.3390/antibiotics10091140

**Published:** 2021-09-21

**Authors:** Janina Noster, Philipp Thelen, Axel Hamprecht

**Affiliations:** 1Department of Medical Microbiology and Virology, Carl von Ossietzky University Oldenburg, 26129 Oldenburg, Germany; janina.noster@uol.de (J.N.); philipp.thelen@uol.de (P.T.); 2Institute for Medical Microbiology and Virology, Klinikum Oldenburg, 26133 Oldenburg, Germany; 3Institute for Medical Microbiology, Immunology and Hygiene, University Hospital, University of Cologne, 50935 Cologne, Germany; 4German Centre for Infection Research, Partner Site Bonn-Cologne, 50937 Cologne, Germany

**Keywords:** multidrug resistance, preanalytical parameters, detection methods, ESBL, carbapenemase, CPE

## Abstract

Multidrug-resistant *Enterobacterales* (MDRE) are an emerging threat to global health, leading to rising health care costs, morbidity and mortality. Multidrug-resistance is commonly caused by different β-lactamases (e.g., ESBLs and carbapenemases), sometimes in combination with other resistance mechanisms (e.g., porin loss, efflux). The continuous spread of MDRE among patients in hospital settings and the healthy population require adjustments in healthcare management and routine diagnostics. Rapid and reliable detection of MDRE infections as well as gastrointestinal colonization is key to guide therapy and infection control measures. However, proper implementation of these strategies requires diagnostic methods with short time-to-result, high sensitivity and specificity. Therefore, research on new techniques and improvement of already established protocols is inevitable. In this review, current methods for detection of MDRE are summarized with focus on culture based and molecular techniques, which are useful for the clinical microbiology laboratory.

## 1. Intestinal Colonization by Multidrug-Resistant Bacteria and the Impact on Global Health

The dissemination of multidrug-resistant (MDR) bacteria is one of the biggest threats to global health [1,2]. In particular, multidrug-resistant *Enterobacterales* (MDRE) are a major cause of hospital-acquired infections, which are associated with high morbidity and mortality as well as rising healthcare costs [3,4,5]. On the other hand, *Enterobacterales* as well as MDRE are part of the normal intestinal microbiota of healthy individuals [6]. The majority of microorganisms in human feces are anaerobic bacteria, but *Enterobacterales*, which account for only 0.01% of 10^13^–10^14^ organisms, are those mainly associated with antibiotic resistance [7]. 

Colonization-rates with MDRE in healthy adults are mostly studied for extended spectrum β-lactamase (ESBL) and AmpC producing *Enterobacterales.* Great variations are observed in different regions, with lower rates in some European countries like Hungary (2.6%) and Sweden (7%) and higher rates in countries like Nepal (9.8%), Mozambique (20%) and Taiwan (41.4%) [8,9,10,11,12]. The main risk factors for new colonization with MDRE among the healthy population are travel to high incidence countries and antibiotic use [9,13]. Risk factors for colonization with MDRE upon hospital admission are previous MDR carriage, travel outside Europe and treatment of gastroesophageal reflux disease and in particular antibiotic treatment within the previous six months [14]. Prevalence of colonization is higher among elderly people in long term care facilities, reflecting antibiotic use, chronic wounds, medical devices, dementia and age as important risk factors. As seen in healthy adults, colonization-rates are highest in Asia with 71.6% for ESBL—producing *Enterobacterales* (ESBLE) and 6.9% for carbapenemase—producing *Enterobacterales* (CPE). In North America, Europe and Oceania, prevalence is 9%, 12.9% and 6% for ESBLE and 5%, 0.2% and 0.4% for CPE, respectively [15]. The prevalence of CPE can vary substantially between different countries on the same continent [16]. In Germany, low CPE prevalence in intensive care units (ICU) has been reported [17], whereas in other European countries like Romania and Italy, CPE are frequently isolated in ICU patients or long-term acute-care facilities with reports of colonization-rates as high as 21.2% and 28.4%, respectively [18,19]. 

Among MDR pathogens, WHO has declared carbapenem- and 3rd generation cephalosporin-resistant *Enterobacterales* as the highest priority for the development of new antibiotics [20]. Surveillance data from the European Centre of Disease Prevention and Control (ECDC) show that the rate of *E. coli* clinical isolates resistant to 3rd generation cephalosporins has been increasing in European countries since 2015 and ranged from 6.2% in Norway to 38.6% in Bulgaria in 2019. Even higher rates have been observed in *Klebsiella pneumoniae* (4.3–75.7%). In parallel, the proportion of *E. coli* and *K. pneumoniae* with resistance to carbapenems increased sharply by 2019 (range 0–1.6% and 0–58.3% for *E. coli* and *K. pneumoniae*, respectively). The prevalence of *E. coli* and *K. pneumoniae* isolates resistant to carbapenems or 3rd generation cephalosporins varies widely among EU/EEA countries, with the overall highest resistance rates being reported from Italy, Bulgaria, Greece and Romania [21].

Resistance to 3rd generation cephalosporins and carbapenems is mostly caused by different β-lactamases, with >7000 different β-lactamases known, and more being published almost daily, illustrating the high mutation and adaptation frequency in different bacteria [22]. β-lactamases are classified according to Ambler (classes A to D) based on primary sequence similarities or according to Bush–Jacoby–Medeiros (group 1 to 3), which is a functional classification depending on biochemical function and susceptibility towards β-lactamase inhibitors [23,24]. Group 1 summarizes all cephalosporinases (Ambler class C) that were originally chromosomally encoded. Group 2 includes all other serine β-lactamases and is composed of several subgroups (Ambler classes A and D). Finally, group 3 combines all metallo-β-lactamases (MBLs, Ambler class B) [24]. 

Extended-spectrum β-lactamases (ESBLs) have been increasingly detected in hospitalized patients and the community since the 1980s and are of great clinical importance. These enzymes are capable of hydrolyzing penicillins, monobactams and 3rd generation cephalosporins. ESBLs belong to Ambler classes A and D in most cases, with those in Ambler class A being mostly sensitive to inhibitors such as clavulanic acid [23]. Relevant examples for β-lactamases in *Enterobacterales* in each class are shown in Figure 1. Comprehensive details on ESBL-types are beyond the scope of this article but have been excellently reviewed elsewhere [23,24,25].

Infections with ESBL-harboring bacteria have been successfully treated with carbapenems, but the massive use of this antibiotic class accelerated the dissemination of the second important group of β-lactamases, the carbapenemases. These enzymes belong to Ambler classes A, B and D and differ in the resistance pattern they induce [26]. Class A includes serine β-lactamases such as *Klebsiella pneumoniae* carbapenemase (KPC-2) [27,28], non-metallo carbapenemase (NMC) [29], imipenemase (IMI) [30] and others. Class B carbapenems are metallo-proteins and are capable of hydrolyzing all antibiotics except monobactams [24]. Well-known examples include NDM-1 [31], VIM [32] and IMP [33]. Oxacillinases and OXA-48 derivates are classified as class D carbapenemases. Their hydrolytic activity against carbapenems and some 3rd generation cephalosporins is lower compared to other carbapenemases, and they are not inhibited by clavulanic acid and tazobactam [34]. 

With the increasing use of antibiotics, the further emergence of MDR *Enterobacterales* seems inevitable. High consumption of antibiotics in health care and livestock as well as increased mobility represent only two accelerators in the process, which is accompanied by an ongoing evolution of resistance plasmids and mobile genetic elements [35,36]. Nevertheless, several intervention strategies are planned or already implemented to reduce the dissemination of MDRE, i.e., by improved infection control, antimicrobial stewardship programs and by decreasing carbapenem use, e.g., by β-lactam-lactamase-inhibitor combinations [37]. In addition, improved detection of MDRE isolates allows rapid isolation of patients colonized or infected with these organisms and may enable a specific antibiotic therapy after determination of the susceptibility patterns. Both steps can reduce spread of resistances, e.g., in hospital settings. However, not only rapid and sensitive detection methods but also pre-analytical parameters affect results and final treatment success.

## 2. Detection and Characterization of MDR *Enterobacterales*

### 2.1. Preanalytical Considerations

Every clinical laboratory must be able to reliably detect MDR bacteria and provide regular reports on the epidemiology of MDRE in order to guide infection control measures. This baseline epidemiological monitoring uses data from clinical samples in most institutions. For high-risk patients, in outbreak situations and in the epidemic setting, the European Society of Clinical Microbiology and Infectious Diseases (ESCMID) as well as the Healthcare Infection Control Practices Advisory Committee and Center for Diseases Control and Prevention (HICPAC/CDC) guidelines recommend active screening cultures (ASC) for the detection of colonization with MDRE, especially for carbapenemase-producing *Enterobacterales* (CPE) [38,39].

An often-overlooked issue which impacts the sensitivity of microbiological cultures and in particular ASC are preanalytical factors including appropriate specimens, the collection site as well as the collection device. Stool is considered the gold standard specimen for studying gastrointestinal microbiota and detection of MDRE colonization [40]. Because of the inconvenient sample collection and processing of stool samples, rectal or perianal swabs are the specimen most commonly used for ASC. The ESCMID guideline for the management of multidrug-resistant Gram-negative bacteria (GNB) considers rectal swabs, urine or respiratory secretions as adequate specimen [38]. The HICPAC/CDC guidelines additionally recommend sampling skin lesions, wounds and sputum or endotracheal aspirates, if colonization of the respiratory tract is suspected [39,41]. In a recent study on colonization sites, it has been shown that 14% of ESBL-GNB carriers are not detected by rectal sampling but by sampling of other sites (e.g., urogenital or respiratory). The authors highlight the need to include samples from the site where MDR-GNB were initially detected for follow-up cultures [42]. For the assessment of gastrointestinal colonization, rectal swabs are preferred over perianal swabs as they recover significantly higher quantities of GNB and have a higher sensitivity for the detection of MDRE [43,44]. Additionally, collection devices have an impact on bacterial recovery: Warnke and colleagues tested nylon-flocked, polyurethan-cellular-foam and classic rayon tip swabs for intra-anal and perianal swabbing. The nylon-flocked and polyurethan-cellular-foam tip swabs recovered significantly higher bacterial loads compared to other swabs, thereby increasing sensitivity for MDRE screening [43].

### 2.2. Culture Based Methods

#### 2.2.1. Screening with Selective Chromogenic and Non-Chromogenic Media

Selective media are suitable for screening of patient samples for ESBL- and carbapenemase-producing *Enterobacterales*. Frequently, agars are supplemented with chromogens which allow presumptive species identification using species-specific enzymes, namely, β-galactosidase, β-glucuronidase and deaminase [45]. 

Several agars are available for ESBL detection, including CHROMagar ESBL (CHROMagar, Paris, France), chromID ESBL agar (bioMérieux, I’Etoile, France), ESBL chromogenic agar (Condalab, Madrid, Spain), chromogenic ESBL agar (SGL, Corby, UK), ESBL ChromoSelect Agar (Merck, Darmstadt, Germany), CHROMagar ^TM^ ESBL (Mast Group, Bootle, UK), Chromatic ESBL agar (Liofilchem, Roseto degli Abruzzi, Italy), Brilliance ESBL agar (Oxoid, Basingstoke, UK), BLSE agar (AES Laboratoire, Combourg, France), and others (Table 1). Most chromogenic ESBL screening agars contain an extended-spectrum cephalosporin (e.g., cefpodoxim) and a mixture of antibiotics to inhibit growth of non-ESBL-producing bacteria. Some agars (i.e., CHROMagar ESBL, chromID ESBL and Brilliance ESBL) also contain additional AmpC-inhibitors [45]. 

Many of these products have been evaluated in different studies. ChromID ESBL agar (bioMérieux, Marcy-I’Etoile, France) and BLSE agar medium (AES Laboratoire, Combourg, France) were challenged with various *Enterobacterales* from clinical samples (rectal swabs, urine samples and pulmonary aspirates). After 24 h, incubation sensitivity was 88% and 85% for chromID and BLSE agar, respectively. Further 24 h incubation increased sensitivity of chromID to 94% but did not affect performance of BLSE agar. The reason for these differences was likely the choice of cefpodoxime in chromID ESBL but cefotaxime or ceftazidime in BLSE agar. False-positive results were obtained for isolates overproducing a chromosomal cephalosporinase or penicillinase [98]. Sensitivity values comparable to ChromID ESBL were obtained with Brilliance ESBL agar (Oxoid, Basingstoke, UK) [99]. Comparison of CHROMagar, chromID, Brilliance ESBL and BD Drigalski supplemented with ceftazidime demonstrated that sensitivity for ESBL detection ranged from 97.2% to 98.6% (CHROMagar 98.3%, ChromID 97.5%, Brilliance ESBL 98.6%, BD Drigalski 97.2%) and specificity from 57.9% (Brilliance ESBL) to 72.9% (chromID) [100]. 

In additional to ESBL agars, chromogenic agars for the detection of CPE have become commercially available in the last decade. In a recent study, seven commercially available screening media and in-house agars were investigated [47]. The authors challenged Brilliance CRE (Oxoid, Basingstoke, UK), Chromatic CRE (Liofilchem, Roseto degli Abruzzi, Italy), chromID CARBA and chromID OXA-48 (bioMérieux, Marcy-I’Etoile, France) and McConkey agar supplemented with ertapenem and cloxacillin with a total of 69 carbapenemase-producing isolates and 40 control strains without carbapenemase-production. In addition, three ESBL agars were assessed for CPE detection, namely Chromatic ESBL (Liofilchem, Roseto degli Abruzzi, Italy), chromID ESBL (bioMérieux, Marcy-I’Etoile, France) and Brilliance ESBL (Oxoid, Basingstoke, UK). The authors demonstrated that ESBL screening media are not suitable for detection of OXA-48-like-producing *Enterobacterales* with low cephalosporin MICs. High specificity and sensitivity for CPE detection was recorded for Brilliance CRE and the in-house agars, whereas chromID CARBA failed to detect 15% of all CPE and 8/20 OXA-48-producing isolates. Because detection of OXA-48-producing *Enterobacterales* is difficult, the use of the specific chromID OXA-48 agar is beneficial according to the authors, especially in outbreaks of OXA-48 CPE [45]. The results are partly comparable to a study by Simner et al. [48]. Again, the authors demonstrated that CPE detection using ESBL screening media is not useful due to reduced sensitivity and specificity (tested for Colorex C3Gr (EO Labs, Bonnybridge, Scotland) and Brilliance ESBL). In contrast to the previously mentioned study, the authors determined the highest sensitivity and specificity for chromID CARBA, followed by Colorex KPC (EO Labs, Bonnybridge, Scotland) and Brilliance CRE but showed also poorer detection of OXA-48 producers [48] (Table 1). 

Pre-enrichment of screening cultures has been proposed as an additional means to increase the sensitivity of MDRE detection. An unselective or semi-selective broth (e.g., MacConkey broth or tryptic soy broth ± antibiotics) is inoculated with the screening sample and incubated overnight before an aliquot is spread on a selective agar. In several studies, pre-enrichment enhanced the detection rate of ESBL or 3rd generation cephalosporin resistant *Enterobacterales* by 21% to 32% [40,101,102,103]. For surveillance studies and in outbreak situations, the improved performance is advantageous. However, this technique is labor-intensive and increases the turn-around-time (TAT) by one day.

Obviously, the performance of the different chromogenic screening media may vary depending on the type of β-lactamase and the analyzed organism. Therefore, the proper choice of chromogenic agar must be adapted to the specific objectives and the epidemiological landscape of each medical center. While the ease of use and experience in the microbiology laboratories are good arguments in favor of chromogenic agar, this technique has some drawbacks. These include a relatively high frequency of unspecific growth on the plates resulting in considerable subsequent work-up, problems in the detection of some β-lactamases (e.g., OXA-48-like) and lack of information on the type of β-lactamase present. Therefore, additional work-up is required to assess susceptibility and identify the ESBL or carbapenemase variant.

#### 2.2.2. Susceptibility Testing of MDRE

Susceptibility can be assessed by various assays, e.g., disc diffusion according to EUCAST or CLSI, semi-automated susceptibility with commercial assays (e.g., Vitek2 (bioMérieux, Marcy-I’Etoile, France), Phoenix (BD Diagnostics, Sparks, MD, USA) or WalkAway (Beckman-Coulter, Brea, CA, USA)) or broth microdilution.

The sensitivity and specificity of commercially available semi-automatic systems varies widely for detection of ESBLs and carbapenemases [104,105,106,107,108,109]. Screening cut-offs have been defined by EUCAST for isolates, which should be further characterized for ESBL- or carbapenemase production [110].

#### 2.2.3. Confirmation Tests

If the results of susceptibility testing show an ESBL- or CPE-phenotype, confirmation tests are necessary to rule out resistance by other mechanisms (e.g., porin loss in combination with other β-lactamases). A great variety of confirmation tests are available, differing in performance, costs and TAT (Table 1).

#### 2.2.4. Disc Diffusion Assays for Detection of ESBL and Carbapenemase Production

In disc diffusion assays, an antibiotics disc is placed on an agar plate inoculated with the test strains. If the test strain is susceptible to the antibiotic, a growth inhibition zone becomes visible after incubation (Figure 2A). For confirmation of screening results and identification of underlying resistance mechanisms, inhibitor-based disc tests are commonly performed, which are distinguished as double-disc synergy test (DDST) and combined disc test (CDT). In the latter assay, a disc containing an antibiotic and a disc containing the antibiotic and a β-lactamase inhibitor are used. If inhibition diameters are increased by the inhibitor by a certain threshold (e.g., 5 mm for CLSI ESBL test) or more or have a 50% change in zone diameter, ESBL production is indicated [111] (Figure 2C). In contrast, in DDST assays, antibiotic and inhibitor are separated in two individual discs which are placed close to each other (also called disc approximation test, Figure 2D). If a β-lactamase is present, an enlargement of the inhibition zones between the two discs is observed. For ESBL detection, cephalosporins and clavulanate are most often employed [112]. 

Similarly, inhibitor-based tests can be used for carbapenemase identification. Carbapenemase inhibitors include boronic acid derivates (KPC and Ambler class A carbapenemase and AmpC β-lactamase inhibition), cloxacillin (AmpC inhibition) and zinc chelators such as EDTA and dipicolinic acid (MBL inhibition) [113,114,115]. OXA-48-like producers are difficult to identify by disc tests as there is no specific inhibitor available for class D carbapenemases. As a work-around, resistance to temocillin and piperacillin-tazobactam can be used as an indicator but is not specific for OXA-48-like [116,117]. 

The faropenem disc test is a highly sensitive screening method for CPE [50,118]. While most CPE show no inhibition zone around the faropenem disc, OXA-48-like *Enterobacterales* often have a double inhibition zone, which is highly specific. The combination of temocillin and faropenem discs increases specificity for OXA-48-like carbapenemase-producing *Enterobacterales* [119].

Both CDT and DDST assays are commercially available for screening ESBL- and carbapenemase-producing *Enterobacterales.* Examples of ESBL tests are Liofilchem ESBL disc kits (Liofilchem, Roseto degli Abruzzi, Italy) and Mastdiscs ESBL detection set (MAST Group, Bootle, UK), while tests for carbapenemase detection are more numerous and include Mastdiscs Carbapenemase Detection Set, Combi Carba Plus (both MAST Group), KPC/MBL & OXA-48 Confirm Kits (Rosco Diagnostica, Taastrup, Denmark) and KPC&MBL&OXA-48 disc kit (Liofilchem, Roseto degli Abruzzi, Italy) [49,50,51]. The latter three tests have been recently evaluated. The test from Rosco diagnostica (ROS) and the Combi Carba plus assay from MAST group (MAST-CDT) detected 86% of carbapenemase-producing strains, while the assay developed by Liofilchem (LIO-CDT) detected 96%. However, false-positive results occurred more frequently with LIO-CDT (6 of 47 carbapenemase negative isolates) than with ROS and MAST-CDT (1 of 47 negative strains each). Classification according to Ambler was correct in 85%, 84% and 96% of CPE for MAST-CDT, ROS-CDT, and LIO-CDT, respectively. Identification of carbapenemases was highly dependent on the carbapenemase-subgroup for all tests. While MAST-CDT and LIO-CDT detected 94% of all class B enzymes, ROS-CDT was more successful in class B detection and LIO-CDT in class D detection (both 100%).

#### 2.2.5. Modified Hodge Test

The modified Hodge test (MHT) is an additional test for the detection of carbapenemases and based on the inactivation of a carbapenem. A susceptible strain (usually *E. coli ATCC 25922*) is inoculated onto an agar plate, and a carbapenem disc is placed in the center. The test strains as well as control strains are streaked in lines from the center to the periphery of the plate. If the test strain produces a carbapenemase, the carbapenem is inactivated and the susceptible *E. coli* strain grows alongside the test strain, which can be observed by cloverleaf-like indentations (Figure 2B). The performance of MHT is lower than that of other inactivation tests, especially for Ambler class B carbapenemases such as NDM-1 (sensitivity 50% without addition of zinc sulfate). Additionally, a low specificity and high numbers of false-positive results are observed in isolates overproducing AmpC or ESBL [52]. Therefore, the CLSI no longer recommends this assay since 2018 [120].

#### 2.2.6. β-Lactam Inactivation Assay (e.g., CIM)

In β-lactam inactivation assays, a susceptible indicator strain is inoculated onto an agar plate and challenged with an antibiotic disc previously incubated in a suspension of the test strain. If the test strain produces a β-lactamase capable of hydrolyzing the drug of interest, the zone of inhibition of the susceptible strain will decrease compared to the control with an untreated antibiotic disc (Figure 2E). The carbapenem inactivation method (CIM) is currently recommended by EUCAST and CLSI for detection of carbapenemases and has a high sensitivity and specificity [110,121]. 

The original protocol was used for the detection of carbapenemase-producing bacteria and recommended suspending the test strain in 400 µl of water, adding an antibiotic disc, followed by 2 h of incubation at 37 °C [53]. While the principle of this assay is applicable for the detection of ESBLs and carbapenemases in Gram-negative bacteria including nonfermenters, it is most commonly used for carbapenemase detection in *Enterobacterales*. Both cultured bacteria and blood-culture fluid can be used for the β-lactam inactivation assay [56,122].

The first established CIM protocol was modified in several parameters to improve the performance. In the so-called modified CIM-test (mCIM), the test strain is suspended in tryptic soy broth instead of water, and incubation time of the antibiotic disc in the bacterial suspension was increased from 2 h to 4 h, which improves sensitivity. If the test strains are additionally incubated with the antibiotic disc and either EDTA (eCIM) or EDTA and phenylboronic acid (CIMPlus), the putative Ambler class of the isolate can be identified [123,124]. The addition of zinc sulfate (zCIM test) to tryptic soy broth has been shown to improve the detection of weakly expressed metallo-β-lactamases, e.g., VIM or NDM (Ambler class B) [50,55]. To reduce the time-to-result, Jing et al. developed the rapid carbapenemase detection method (rCDM). In this modified protocol, test strains grown overnight on blood agar are smeared on an imipenem disc and then placed on an agar plate inoculated with a susceptible indicator strain. This test can already be read after an incubation of 5–6 h [57]. 

All CIM assays perform well with high specificity and sensitivity (81–100% and 97–100%, respectively), but false-positive results can occur in strains with reduced porin expression and overexpression of AmpC β-lactamase or ESBL. In addition, detection of strains with low carbapenemase-activity remains challenging.

#### 2.2.7. Colorimetric Assays

While CIM assays still have a time-to-result of 5 to 20 h, colorimetric assays have been developed for rapid detection of ESBL and carbapenemases. The ESBL-NDP test is employed for detection of ESBL activity. Lysed bacteria are incubated in a medium containing red phenol as pH indicator and supplemented with either cefotaxime, cefotaxime and tazobactam or without antibiotics. Hydrolysis of cefotaxime by ESBL leads to the formation of carboxylic acid which induces a color change from red to yellow. In the case of ESBLs, this reaction can be inhibited by tazobactam (Figure 3). The test was also challenged with spiked blood samples and urine and demonstrated good sensitivity and specificity [125]. The principle of the NDP test (also called NP-test) underlies the commercially available Rapid ESBL Screen kit 98022 (Rosco Diagnostica, Taastrup, Denmark) [126]. Other colorimetric test kits include the β-Lacta test (Bio-Rad, Marnes-La-Coquette, France), but this assay is used for the detection of 3rd generation cephalosporin-resistance and cannot distinguish between carbapenemases and ESBLs [127]. 

Comparable to the detection of ESBL enzymes by pH decrease, the Carba NP test has been developed for detection of carbapenemases. The hydrolysis of imipenem by a carbapenemase leads to an acidification of a phenol red solution, resulting in color change from red to orange or yellow [128]. The follow-up version Carba NP test II again includes inhibitors (tazobactam and EDTA) to differentiate between Ambler classes [129]. Modified variants of this test are suitable for the detection of CPE in blood cultures [56,58]. A disadvantage of the colorimetric Carba NP assays is the high number of false-negative results for *Enterobacterales* and *Pseudomonas* spp. with low hydrolytic activity, such as OXA-48 [130]. In a variation of the assay, this already rapid test (2 h time-to-result) was modified for even faster results by omitting the lysis step (CNPt-direct test) and including the detergent Triton X-100 instead [65]. Improved OXA-48 detection was achieved by Ma et al. who developed the substrate CARBA-H, which provides better enzyme-substrate interaction for OXA-48 and shows strong fluorescence upon cleavage [66].

Due to the increasing number of CPE, a variety of detection kits are commercially available, which include the β-CARBA assay (Biorad, Hercules, CA, USA), NeoRapid CARB (Rosco Diagnostica, Taastrup, Denmark) and CARBA PAcE (Mast Group, Bootle, UK) (Table 1). These tests have been validated in several studies and showed overall good performance. Poor detection of metallo-carbapenemases could be partly improved by use of bacterial material from blood agar instead of MH-agar or supplementation of the growth medium with zinc sulfate [55,56,60,62].

Colorimetric tests provide a rapid indication if the test strain is an ESBL- or carbapenemase producer and tests with inhibitors can identify which Ambler class the enzymes belong to. However, precise identification of the type of ESBL or carbapenemase cannot be achieved using cultural methods and requires molecular techniques.

### 2.3. Molecular Methods

Molecular methods have the highest sensitivity and specificity for the detection of resistance genes and can be applied to cultured isolates or directly to clinical specimens. They can provide accurate information on the type of ESBL enzyme or carbapenemase and have a relatively short time-to-result. Although molecular methods have the disadvantage of requiring qualified personnel and incur high costs for technical equipment and consumables, the number of laboratories using these techniques is growing. This is due in part to the continuously increasing number of different methods and commercially available assays (Table 1). In particular, ESBL and carbapenemase screenings at hospital admission benefit from rapid results without prior cultivation steps.

#### 2.3.1. PCR, RT-PCR and Microarray Techniques

For simultaneous detection of multiple target genes, multiplex PCR and microarrays are the methods of choice. These assays provide a more comprehensive picture of the resistance landscape of an isolate [131], since many isolates harbor multiple resistance genes [132]. A major advantage of resistance gene detection is that low expression levels and catalytical activities do not impact the result, allowing detection of β-lactamases with low activity [131]. A disadvantage is that only genes which are targeted in the assay can be detected. Therefore, multiplex PCR and microarray assays have to be continuously improved and expanded with the emergence of new resistance genes. 

Several in-house multiplex PCR assays have been published [132,133,134,135]. Today, commercially available PCR-assays are available and frequently preferred over in-house tests. In order to detect patients harboring MDR bacteria at hospital admission, it is useful to select assays that can be performed directly from rectal swabs. One example is the ESBL ELITe MGB Kit (ELITechGroup, Puteaux, France). It is a multiplex real-time PCR assay for the detection of CTX-M-genes and can be used on blood cultures or rectal swabs. The assay performed well, showing 100% sensitivity and 96.6% specificity [85]. Another multiplex real-time PCR assay specific for detection of ESBL-producing bacteria is the Check-Direct ESBL Screen for BD MAX (Check-Points Health BV, Wageningen, Netherlands) which can detect the ESBL gene families CTX-M-1, CTX-M-2, CTX-M-9 and SHV-ESBL. However, sensitivity of this assay was 95.2% but only evaluated on a small number of ESBL positive specimens (*n* = 21). Specificity was slightly higher at 97.6% and modified cut-off values even increased specificity to 98.8% [136]. 

The number of commercially available PCR-based assays for detection of carbapenemase-producing *Enterobacterales* is even higher (Table 1). Many of these assays are also suitable for direct detection of CPE from rectal swabs. Checkpoint developed the BD MAX Check-Points CPO-assay for detection of KPC, VIM/IMP, NDM and OXA-48 carbapenemase-producers. In one study, this assay was challenged using 128 rectal swabs, among others, and demonstrated a sensitivity and specificity of 92.8% and 97.8%, respectively. Nevertheless, rare IMP and OXA-48-like producers (IMP-11, IMP-13, IMP-14) were missed by this assay. Results were obtained within 2.5 h for 12 samples at a time [83]. In another study, comparable results were obtained with the BD MAX Check-Points CPO-assay for rectal swabs, which achieved a sensitivity of 96.6–100% and a specificity of 98.3–100%, depending on the type of carbapenemase tested [137]. Another technique is used in the Check-MDR CT103 XL assay (Check Points, Wageningen, The Netherlands), which combines a targeted PCR with array methods for detection. It detects 11 carbapenemases, 19 ESBL groups and subgroups, and additional minor ESBLs, AmpCs and MCR [138]. The predecessor Check-MDR CT103 performed well in the study by Cuzon et al., with sensitivity and specificity of 95–100% and 100%, respectively [84]. Other tests for CPE detection from rectal swabs include the Xpert Carba-R assay for the GeneXpert system (Cepheid, Sunnyvale, California [139,140,141]), the CRE ELITe MGB Kit (ELITechGroup, Puteaux, France [85]), GenePOC/Revogene Carba C assay (Meridian Bioscience, Cincinnati, Ohio [46]), and assays developed by MOBIDIAG for the Amplidiag^®^ Easy system, namely, Amplidiag CarbaR + VRE and Amplidiag CarbaR-MCR assays (Mobidiag Ltd., Espoo, Finland [86,87]).

#### 2.3.2. Loop-Mediated Isothermal Amplification Assay (LAMP)

LAMP can circumvent the disadvantages of PCR assays, namely the high costs and negative influence of compounds present in clinical samples which can inhibit DNA polymerase activity. Instead of repetitive temperature cycles like in PCR, only isothermal conditions are required. Good results were obtained with the eazyplex SuperBug CRE system (Amplex Biosystems GmbH, Giessen, Germany), which can detect ESBLs of the CTX-M-1 and CTX-M-9 group, and the carbapenemases KPC, VIM, NDM, OXA-48 and OXA-181. The assay showed 100% concordant results compared to PCR based assays when using cultured *E. coli* isolates [142]. Comparable results were reported for eazyplex SuperBug CRE on *Klebsiella* spp. and *Pseudomonas aeruginosa* isolates [143] and on cultured *Enterobacterales* and *Pseudomonas* isolates [82,144]. The assay was also able to detect ESBL-encoding genes in urine samples with high sensitivity (100%) and specificity (97.8%) [145]. A drawback is the lack of detection of IMP-carbapenemases and the small number of studies with clinical specimens, such as rectal swabs.

Molecular methods are advantageous due to their high sensitivity and specificity, but not all diagnostic laboratories are equipped with the appropriate devices and trained personnel. In addition, molecular methods are expensive compared to phenotypic assays. Further developments in the field of molecular diagnostics could circumvent these drawbacks and lead to a more widespread implementation of these techniques in routine laboratories.

### 2.4. Further Methods

#### 2.4.1. Immunochromatographic Test (ICT)

Most ICTs in the microbiological laboratory are lateral flow double-antibody sandwich assays. After application of the analyte to the sample pad, the antigens (e.g., β-lactamases) migrate along the nitrocellulose membrane by capillary flow. They form a complex with dye-labeled antibodies (e.g., gold particles, latex microspheres) on the conjugate pad and are captured by immobilized antibodies on a nitrocellulose membrane at the test line resulting in a visible band. The control line consists of immobilized anti-IgG antibodies which capture the excess dye-labeled antibodies, thereby serving as a control for the capillary flow (Figure 4). The main advantage of ICTs is that they are easy to use, relatively cheap, do not require additional and costly analyzers and provide a very short time-to-result.

The NG-Test CTX-MULTI (NG biotech, Guipry, France) was developed to detect group CTX-M-1, -2, -8, -9, -25 producing *Enterobacterales* from bacterial colonies and positive blood cultures [146]. The authors point out that in settings with high 3rd generation cephalosporin resistance due to other resistance mechanisms such as plasmid-mediated AmpC, this test will lack sensitivity. Addressing this restriction, Moguet et al. developed a functional ICT using an incubation step of a bacterial colony with cefotaxime and anti-cefotaxime antibodies to detect expanded cephalosporinase activity regardless of its conferring enzyme (unpublished data).

For carbapenemases, different tests are commercially available and have been extensively studied. The NG-Test CARBA-5 (NG biotech, Guipry, France) detects the most prevalent carbapenemases KPC, OXA-48-like, VIM, IMP, and NDM from bacterial colonies and directly from positive blood cultures with high sensitivity and specificity (84.2–99.2% and 95.3–100%, respectively), (Table 1) [55,69,71,72]. The RESIST-5 O.O.K.N.V. as well as its predecessors RESIST-4 and RESIST-3 (Coris BioConcept, Gembloux, Belgium) can be performed from bacterial colonies as well as directly from positive blood cultures. They detect and differentiate between OXA-48-like, OXA-163 (only RESIST-5), KPC, NDM, and VIM (only RESIST-4 and -5) with high sensitivity and specificity (84.2–100% and 100%, respectively) [55,67,68,69,147]. While the performance for KPC and OXA-48-like carbapenemases is excellent (sensitivity/specificity 100% in most studies), lower sensitivity has been reported for some Ambler class B β-lactamases especially in *Proteus* spp. and from positive blood cultures [67,147,148]. This can be overcome by adding zinc to the incubation protocol or working with zinc supplemented agar plates [67,149].

Despite the numerous advantages of ICTs, they share the same problems with molecular methods, e.g., only the carbapenemases included in the test design can be detected, and rare carbapenemases are usually missed. To overcome this restriction, Baeza and colleagues proposed an algorithm applying immunochromatographic assays in combination with a subsequent zCIM test for the detection of common and rare carbapenemases (Figure 5). Reaching a sensitivity of 99.3% on a large collection of clinical isolates with common and rare carbapenemases, this algorithm provides a robust and cost-effective tool for carbapenemase confirmation, including uncommon variants [55].

Boattini and colleagues proposed a fast-track workflow for *Enterobacterales* from positive blood cultures combining the NG-Test CTX-M MULTI and the NG-Test Carba 5 assay after identification of *Enterobacterales* using the MBT Sepsityper IVD KIT (Bruker DALTONICS, Bremen, Germany). Analyzing a total of 236 episodes of *Enterobacterales* blood stream infections, a good agreement with conventional phenotypic results was recorded. Time-to-result (defined starting point was the processing of positive BC) for this fast-track workflow was 42 min compared to 38 h for the conventional workflow (identification and AST from overnight cultures) [150]. In practice this means that that therapeutic and antibiotic stewardship interventions can be implemented one to two days earlier with this new protocol, which highlights the potential of ICT used in innovative workflows to accelerate clinical decision making.

#### 2.4.2. Electrochemical Assays

Another way to detect resistance to carbapenems or 3rd generation cephalosporins with little equipment is by employing electrochemical assays. They use disposable sensors with screen printed carbon electrodes in combination with small electronical measuring devices.

Bogaerts and colleagues developed the BYG Carba test for the rapid confirmation of carbapenemase activity. The test measures variations of conductivity on an electro-sensing polymer coated electrode conferred by imipenem hydrolysis. These variations are analyzed in real-time by a portable reader [80]. The test was evaluated with a set of 1181 *Enterobacterales* and showed a good sensitivity and specificity for the confirmation of carbapenemases from bacterial colonies (96.3% and 99.7%, respectively) [151].

Rochelet and colleagues described a voltammetric assay for the detection of ESBLs. Bacterial isolates are incubated together with nitrocefin, and subsequently, the hydrolysis of nitrocefin into an electroactive product is detected by a voltammetric measurement on a disposable carbon screen-printed sensor [152].

The successor of this principle is the commercially available BL-RED test (Coris BioConcept, Gembloux, Belgium) which is designed for the detection 3rd generation cephalosporin hydrolysis from bacterial culture or directly from positive blood cultures. Durand and colleagues challenged the test with a set of 150 *Enterobacterales* spiked into blood cultures and reported a good detection of 3rd generation cephalosporin resistance conferred by Ambler class A β-lactamases (sensitivity 83.3% and specificity 99.1%), but the test failed to detect 3rd generation cephalosporin resistance conferred by Ambler classes B, C and D β-lactamases [81].

#### 2.4.3. MALDI-TOF MS

MALDI-TOF MS was a revolution in the field regarding species identification. It is increasingly embraced for the detection of antimicrobial resistance (AMR), and this application will likely become an essential part of the routine laboratory in the near future [153,154]. Regarding the detection of MDR *Enterobacterales,* there are three promising approaches for MALDI-TOF MS application available: The detection of β-lactamase activity, estimating the effect of antibiotics on the growth of bacteria, and the direct detection of biomarkers (e.g., enzymes or target modifications) associated with AMR (Table 1).

The detection of β-lactamase activity using MALDI-TOF is already available for IVD use in Europe. The principle is to incubate the bacterial isolate in question with an antibiotic. Hydrolysis of the β-lactam antibiotic results in a mass shift of the corresponding antibiotic within the spectrum obtained by MALDI-TOF [155,156]. The MBT STAR-Carba and the STAR-Cepha IVD Kits (Bruker DALTONICS, Bremen, Germany) are two commercially available functional tests for the detection of carbapenem resistance and resistance against 3rd generation cephalosporins with high sensitivity and specificity (98–100% and 97–100%, respectively) [74,75,76,77]. The downside of this approach is that it only detects resistance conferred by hydrolysis of the target antibiotic and therefore yields false-negative results for resistance conferred by other mechanisms, e.g., drug efflux or porin loss.

Another approach for MALDI-TOF MS is to use it as a sensitive instrument for growth detection in presence of defined antibiotic concentrations. For this purpose, Idelevich and colleagues developed the direct-on-target microdroplet growth assay (DOT-MGA) [78]. In this assay a defined bacterial inoculum is incubated together with an antibiotic at the clinical breakpoint concentration directly on a MALDI-TOF MS target. These on-target-microdroplets are incubated in a humidity chamber and analyzed after drying with the MALDI Biotyper (Bruker DALTONICS, Bremen, Germany). Species identification on a target with the antibiotic (score ≥ 1.7) is interpreted as non-susceptible whereas missing species identification (score ≤ 1.7) is interpreted as susceptible to the antibiotic. This technique was shown to have an excellent sensitivity and specificity (100%) for the detection of meropenem resistance in *K. pneumoniae* after 4 h incubation from culture plates [78] as well as directly from positive blood cultures (sensitivity 91.7%, specificity 100%) [79]. The technique was successfully expanded to detect ESBL and AmpC β-lactamases in *Enterobacterales* [157]. While this application is still for research-use-only (RUO), it is a very promising approach for the detection of antimicrobial resistance in *Enterobacterales* as it uses an easy-to-follow protocol, an instrument that is available in many microbiological laboratories and has an acceptable TAT. Moreover, it is independent of the resistance mechanisms and can be applied to any combination of antibiotic and bacterial species.

Another technique is the direct detection of biomarkers of antimicrobial resistance in mass spectra. Using this approach, the same mass spectra are used for species identification and resistance detection. This could significantly shorten laboratory workflows and is already established for some resistance markers, e.g., for some MRSA strains and for carbapenem resistance in *Bacteroides fragilis_cfiA_*
_pos._ [158,159]. The detection of mass spectrometry peaks of some β-lactamases such as TEM-1 [160] and KPC-2 [161,162] was shown to distinguish reliably between resistant and susceptible isolates. Cordavana and colleagues designed an algorithm integrated into the MALDI Biotyper System (Bruker DALTONICS, Bremen, Germany) that enables the automated detection of KPC harboring *Klebsiella pneumoniae* from cultured colonies as well as directly from positive blood cultures during the routine identification process (sensitivity 85.1%; specificity 100%) [163]. The protocols for *Enterobacterales* are not yet validated for routine laboratory use but are promising approaches for the near future. For now, biomarkers allow the detection of a few selected resistance determinants, and therefore, confirmation of negative results by phenotypic AST methods is still needed. With better and broader spectra libraries, further resistance mechanisms can likely be detected in the future. Accelerating bioinformatic tools like machine learning will play a crucial role in the construction of those libraries [153].

## 3. Summary and Future Perspectives

Patients colonized or infected by MDRE can today be more rapidly and reliably identified using assays with shorter turn-around-time and improved sensitivity.

Culture based approaches rely on the cultivation on selective or unselective agars, subsequent susceptibility testing and identification of resistance mechanisms. This workflow will likely remain the standard in the medical microbiology laboratory. This approach permits the subsequent characterization of isolates, e.g., by phenotypic methods or molecular methods including next generation sequencing (NGS). For patient screening using the classical culture approach, we recommend using an ESBL in combination with a CPE agar, as not all CPE can be detected by ESBL screening agars. The use of an enrichment broth is optional but will further improve the detection of MDRE.

The traditional culture-based workflow requires about 2.5 days while using molecular techniques directly from screening specimens only requires a few hours. At the time of writing, PCR techniques are still costly when applied routinely for screening purposes from sample material [164]. Since molecular detection of MDRE has demonstrated excellent sensitivity and specificity, PCR is increasingly used in diagnostic practice as costs per test are decreasing and many commercial assays on different molecular platforms have become available. 

Many tests presented in this review have primarily been designed for the confirmation of ESBL and carbapenemase production but have been evaluated for the use directly on clinical specimens, especially blood cultures. The more widespread use of these techniques will accelerate clinical decision making and ultimately improve patient outcomes. Different rapid diagnostic interventions (i.e., rapid ID, rapid AST or the determination of resistance genes) have shown to decrease the time for adequate therapy. For other parameters such as length of stay, overall mortality and cost effectiveness, the picture is less clear. Most studies combine the implementation of rapid diagnostics with antimicrobial stewardship programs making it difficult to assess the genuine impact of new technologies alone [165].

Next generation sequencing (NGS) is increasingly used in microbiological laboratories, mainly in reference institutions and research. The DNA sequence of a bacterial genome can be obtained in a single sequencing run and is very successfully used in the determination of antibiotic resistance genes (ARG) and typing of pathogens in hospital outbreaks. NGS can be employed for in-depth analysis of single isolates but also allows culture-free identification of bacteria and ARG directly from complex samples like stool [166]. New innovative bioinformatics tools not only allow the detection of well-described ARGs but also enable the identification of previously uncharacterized resistance genes from shotgun metagenomic sequencing data of the human microbiota [167]. On the other hand, the genotype of a bacterial isolate does not precisely translate into its resistance phenotype. To date, there is insufficient evidence to infer antibiotic susceptibility from whole genome sequencing data to guide clinical decision making [168].

All these applications make NGS a valuable tool for identification, surveillance and most likely future clinical decision making regarding MDRE. With declining costs for the initial investment in an NGS platform, decreased costs per run and the development of accessible bioinformatics software, this technique will inevitably be incorporated into routine microbiological practice. However, standardization of protocols, quality control issues and bioinformatic capacities are serious obstacles that need to be addressed before a more widespread implementation in clinical microbiology laboratories seems realistic [169].

Overall rapid and reliable detection of MDRE in clinical specimens is a key capacity for every clinical microbiological laboratory. Multidrug resistance in *Enterobacterales* is a rapidly evolving field with a diverse armamentarium of in-house and commercially available tests as well as fascinating future perspectives. 

## Figures and Tables

**Figure 1 antibiotics-10-01140-f001:**
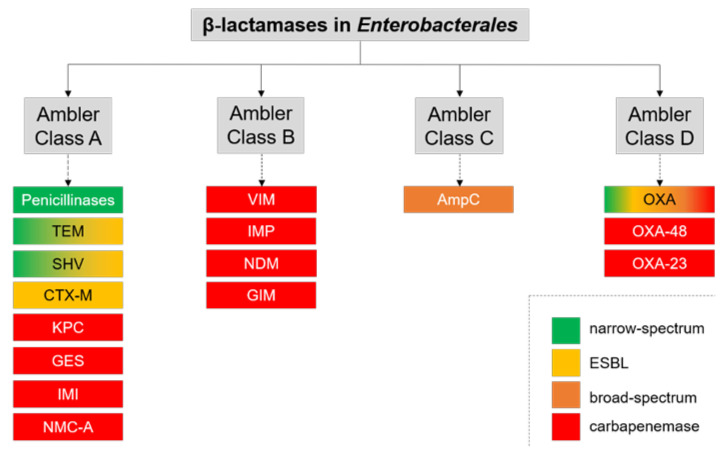
Ambler’s classification with examples of main β-lactamases in *Enterobacterales*.

**Figure 2 antibiotics-10-01140-f002:**
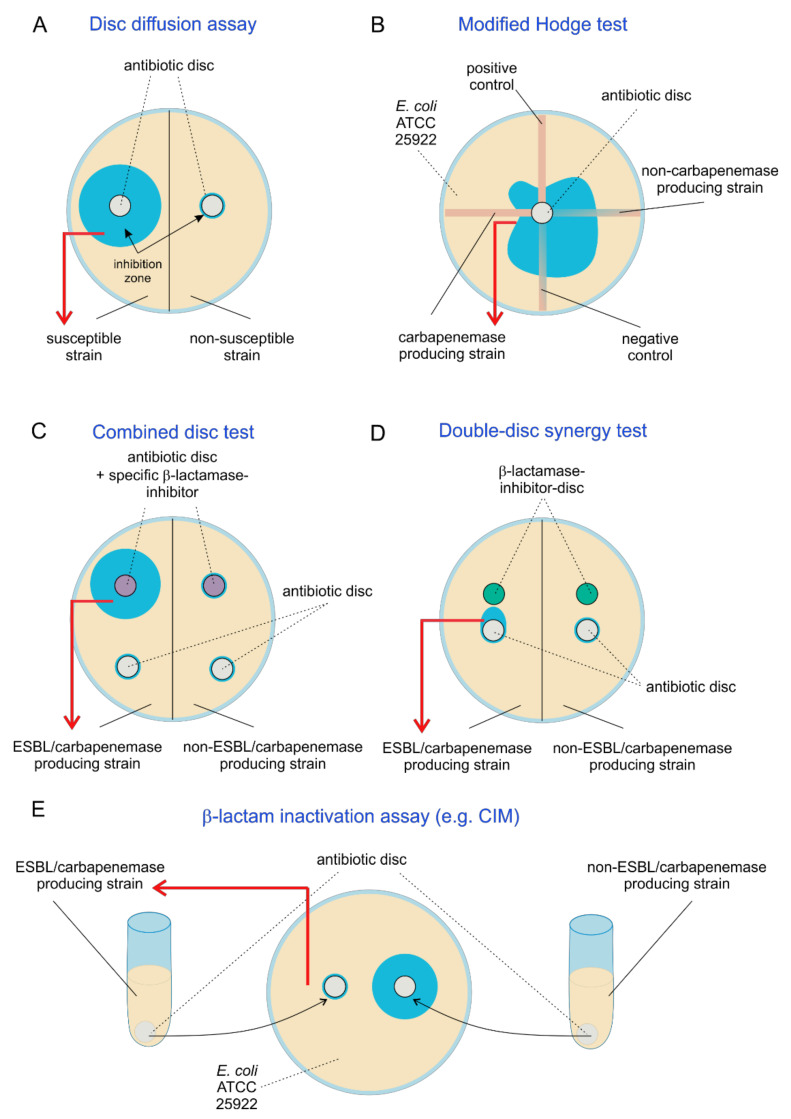
ESBL and carbapenemase detection by disc diffusion tests. Schematic overview of disc diffusion assay variations for detection of ESBL and carbapenemase-producing strains. (**A**) Principle of susceptibility testing by disc diffusion assay. (**B**) Modified Hodge test. (**C**) Combined disc test. (**D**) Double-disc synergy test. (**E**) β-lactam inactivation assay (e.g., CIM).

**Figure 3 antibiotics-10-01140-f003:**
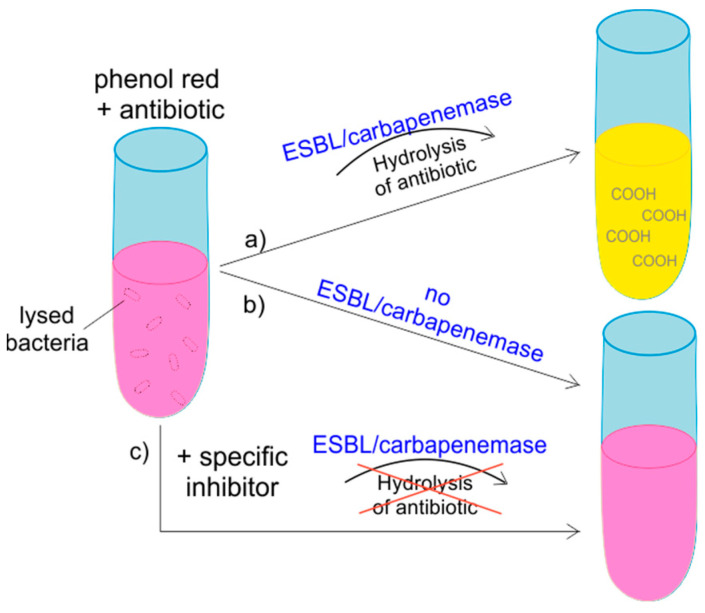
Principle of colorimetric assays for detection of ESBLs and carbapenemases. The test isolate is lysed and incubated in phenol red solution with an antibiotic. (**a**) If the test isolate is positive for ESBL/carbapenemase production, the enzyme hydrolyzes the antibiotic, resulting in a pH shift and a color change from magenta to yellow. (**b**) ESBL/carbapenemase-negative isolates do not induce a change in color. (**c**) Addition of enzyme-specific inhibitors prevents hydrolysis of the antibiotic, and no color change is visible.

**Figure 4 antibiotics-10-01140-f004:**
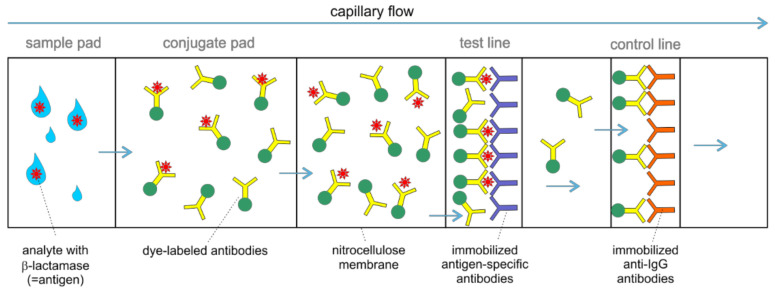
Basic principle of lateral flow assays. Schematic illustration of different zones of commercially available lateral flow assays for detection of, i.e., β-lactamases. Antigens within the sample migrate along the nitrocellulose membrane by capillary flow. Antigens (e.g., β-lactamase) conjugate with specific dye-labeled antibodies and are captured by immobilized antigen-specific antibodies, resulting in the visualization of the test line. Non-conjugated and unbound antibodies migrate further to the control line and are captured by immobilized anti-IgG antibodies, resulting in the visibility of a control line.

**Figure 5 antibiotics-10-01140-f005:**
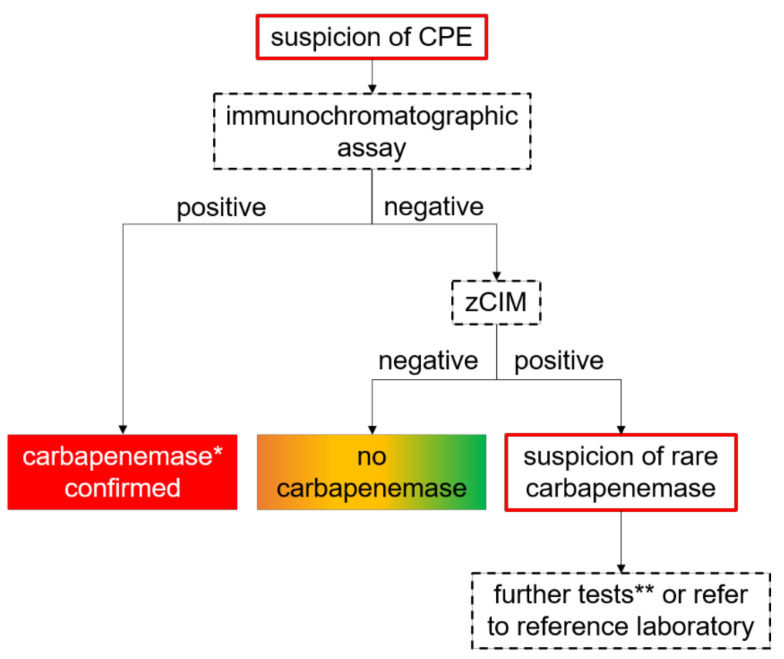
Algorithm for the detection of carbapenemases in routine laboratory as proposed by Baeza et al. * OXA-48-like KPC, VIM, NDM, (IMP). ** Multiplex PCR targeting rare carba-penemases (e.g., IMI, GES, etc.), whole genome sequencing, other tests [61].

**Table 1 antibiotics-10-01140-t001:** Overview of selected versions of carbapenemase detection methods available for *Enterobacterales*. Table modified from [46].

Method	Test	Time to Result	Ease of Interpretation	Target Enzymes/ Genes	Sens. (%)	Spec. (%)	Tested on Clinical Specimens	Reference
Selective chromogenic/non-chromogenic agar	Brilliance CRE (Oxoid)	18–48 h	easy	Classes A, B, D	77.6–98.6	60–87.1	bacterial colonies	[47,48]
	Chromatic CRE (Liofilchem)	18–48 h	easy	Classes A, B, D	94.2	60	bacterial colonies	[47]
	chromID CARBA (bioMérieux)	18–48 h	easy	Classes A, B, D	85.5–89.8	87.5–95	bacterial colonies	[47,48]
	chromID OXA-48 (bioMérieux)	18–48 h	easy	OXA-48	34.8 (all CPE) 100 (OXA-48)	100	bacterial colonies	[47]
	McConkey supplemented with ertapenem, cloxacillin, zinc-sulfate and ticarcillin	24–48 h	easy	Classes A, B, D	97.1	77.5	bacterial colonies	[47]
Disc diffusion and related assays	Mastdiscs ^TM^ Carbapenemase Detection Set (MAST GROUP)	18–24 h	easy	Classes A, B	78	93	bacterial colonies	[49]
	Combi Carba Plus Kit (MAST GROUP)	18 h	easy	Classes A, B, D	86	98	bacterial colonies	[50]
	KPC/MBL & OXA-48 Confirm Kits (Rosco Diagnostica)	18–24 h	subjective	Classes A, B, D	86–98.8	93.1–98	bacterial colonies	[50,51]
	KPC&MBL&OXA-48 disc kit (Liofilchem)	18 h		Classes A, B, D	96	87	bacterial colonies	[50]
	faropenem disc	18 h	easy	Classes A, B, D	99	81	bacterial colonies	[50]
	Modified Hodge Test	18–24 h	subjective	Classes A, B, D	77.4	38.9	bacterial colonies	[52]
	Carbapenem inactivation method (CIM)	18–24 h	easy	Classes A, B, D	n.a.	n.a.	bacterial colonies	[53]
	mCIM	18–24 h	subjective	Classes A, B, D	97	99	bacterial colonies	[54,55]
	zCIM	18–24	easy	Classes A, B, D	97.4–98	97.7–100	bacterial colonies	[50]
	bcCIM	18–24 h	easy	Classes A, B, D	100	100	blood culture fluid	[56]
	rapid carbapenemase detection method (rCDM)	5–6 h	easy	Classes A, B, D	100	99.6	bacterial colonies	[57]
Colorimetric assays	Carba NP test	5–120 min	subjective	Classes A, B, D	97.9	100	spiked blood cultures	[58]
	bcCarba NP test	5–120 min	subjective	Classes A, B, D	99	95.1	blood culture fluid	[56]
	Neo-Rapid CARB Screen (Rosco Diagnostica)	15–120 min	subjective	Classes A, B, D	89.5	70.9	blood culture fluid, urine	[59]
	Neo-Rapid CARB from PBC	90–120 min	subjective	Classes A, B, D	99	91.4	blood culture fluid	[56]
	Rapidec carba NP test (bioMérieux)	5–120 min	subjective	Classes A, B, D	99	100	blood culture fluid	[59]
	β-CARBA test (Bio-Rad)	30 min	subjective	Classes A, B, D	64.9–84.9	90–95.6	bacterial colonies	[60,61]
	β-CARBA test from PBC	30 min	subjective	Classes A, B, D	100	95.1	blood culture fluid	[56]
	CARBA PAcE (MAST GROUP)	10 min	subjective	Classes A, B, D	72	91	bacterial colonies	[62]
	Blue Carba test	30–120 min	subjective	Classes A, B, D	100	100	bacterial cultures	[63]
	Rapid Carb Blue kit (Rosco Diagnostica)	15-60 min	subjective	Classes A, B, D	93.3	100	blood culture fluid, urine	[64]
	CNPt-direct test	2 h	subjective	Classes A, B, D	98	100	bacterial colonies	[65]
	Carba-H-assay	2 h	subjective	Classes A, B, D	n.a.	n.a.	bacterial colonies, spiked urine samples	[66]
Immunochromatographic assays	RESIST-3 O.K.N. assays (Coris BioConcept)	20–45 min	easy	KPC, NDM, OXA-48-like	100	100	blood culture fluid	[67]
	RESIST-4 O.K.N.V. assays (Coris BioConcept)	15 min	easy	KPC, NDM, VIM, OXA-48-like	84.2–99.2	100	bacterial colonies	[55,68]
	RESIST-5 O.O.K.N.V. assays (Coris BioConcept)	15 min	easy	KPC, NDM, VIM, OXA-48-like, OXA-163	99.4	100	bacterial colonies	[69]
	NG-Test Carba 5 (NG Biotech) from PBC	30 min	easy	KPC, NDM, VIM, OXA-48-like, IMP	97.7	96.1	blood culture fluid	[70]
	NG-Test Carba 5 (NG Biotech)	15 min	easy	KPC, NDM, VIM, OXA-48-like, IMP	88.2–100	95.3–100	bacterial colonies	[55,69,71,72]
Mass spectrometry	MALDI-TOF MS	4.5 h	complex	Classes A, B, D	100	90	blood culture fluid	[73]
	MBT STAR-Carba IVD Kit (Bruker DALTONICS)	30–60 min	moderate	mass shift of hydrolyzed carbapenem	98–100	97–100	bacterial colonies	[74,75,76,77]
	MBT STAR-Carba IVD Kit (Bruker DALTONICS) from PBC	1 h	moderate	mass shift of hydrolyzed carbapenem	100	100	blood culture fluid	[76]
	Direct-on-target microdroplet growth assay (DOT MGA), RUO	4–6 h	moderate	bacterial growth in presence of defined carbapenem concentration	100	100	bacterial colonies	[78]
	Direct-on-target microdroplet growth assay (DOT MGA), RUO, from PBC	4–6 h	moderate	bacterial growth in presence of defined carbapenem concentration	91.7	100	blood culture fluid	[79]
Electrochemical assay	BYG Carba test (not yet commercially available)	5–30 min	easy	Classes A, B, D	95	100	bacterial colonies	[80]
	BL-RED test (Coris BioConcept)	20 min	easy	3GC resistance by hydrolysis	46.7 (83.8 for detection of class A β-lactamases)	100	blood culture fluid	[81]
Other molecular methods	Eazyplex Superbug CRE (Amplex Diagnostics)	15 min	easy	*bla*_KPC_, *bla*_VIM_, *bla*_NDM_, *bla*_OXA-48-like_	100	100	blood culture fluid, rectal swab, urine	[82]
Multiplex PCR-based assays	BD MAX ^TM^ Check-Points CPO assay (Check Points)	<180 min	easy	*bla*_KPC_, *bla*_VIM_, *bla*_NDM_, *bla*_OXA-48-like_, *bla*_IMP_	97.1	98.8	rectal swab	[83]
	Check-MDR CT103 (Check Points)	6 h	relatively easy	among others *bla*_KPC2,3_, *bla*_NDM1,2,3_, *bla*_OXA-48_, *bla*_OXA-181_, *bla*_VIM1,2,3,4,19_, *bla*_IMP1,4,8,13_	95–100	100	DNA extracted from bacterial colonies	[84]
	CRE ELITe MGB ^®^ kit (ELITechGroup)	<180	easy	*bla*_KPC_, *bla*_VIM_, *bla*_NDM_, *bla*_OXA-48-like_, *bla*_IMP_	100	100	PBC, rectal swab, respiratory sample	[85]
	Amplidiag ^®^ CarbaR + VRE (Mobidiag Ltd.)	<180 min	relatively easy	*bla*_KPC_, *bla*_VIM_, *bla*_NDM_, *bla*_OXA-48-like_, *bla*_IMP, IS*Aba1*-OXA-51_, *bla*_OXA-23_, *bla*_OXA-24/40_, *bla*_OXA-58_	100	99	DNA extracted from stool samples, rectal swab, pure culture	[86]
	Amplidiag ^®^ CarbaR + MCR (Mobidiag Ltd.)	<180 min	relatively easy	*bla*_KPC_, *bla*_VIM_, *bla*_NDM_, *bla*_OXA-48-like_, *bla*_IMP, IS*Aba1*-OXA-51_, *bla*_OXA-23_, *bla*_OXA-24/40_, *bla*_OXA-58_, *bla*_GES-2_, *bla*_GES4 through_*bla*_GES-6_, *bla*_GES-13 through_*bla*_GES-16_, *bla*_GES-18_, *bla*_GES-20/21_, *bla*_GES-24_	92–100	86–100	DNA extracted from stool samples, rectal swab, pure culture	[87]
	Xpert-Carba-R assay ^®^ (Cepheid)	50 min	easy	*bla*_KPC_, *bla*_VIM_, *bla*_NDM_, *bla*_OXA-48-like_, *bla*_IMP_	97.8	95.3	rectal swab	[88]
	GenePOC/Revogene Carba C assay ^®^ (Meridian)	70 min	easy	*bla*_KPC_, *bla*_VIM_, *bla*_NDM_, *bla*_OXA-48-like_, *bla*_IMP_	100	100	bacterial colonies	[46]
	Verigene BC-GN (Nanosphere)	2–2.5 h	relatively easy	*bla*_KPC_, *bla*_VIM_, *bla*_NDM_, *bla*_OXA-48-like_, *bla*_IMP_, *bla*_OXA-23_, *bla*_OXA-24/40_, *bla*_OXA-58_	100	100	blood culture fluid	[89]
	Biofire ^®^ Filmarray ^®^ Blood Culture Identification (BCID) Panel (bioMérieux)	1 h	easy	*bla*_KPC_, *bla*_VIM_, *bla*_NDM_, *bla*_OXA-48-like_, *bla*_IMP_	n.a.	n.a.	blood culture fluid	[90]
	Biofire ^®^ Filmarray ^®^ Pneumonia plus Panel (bioMérieux)	1 h	easy	*bla*_KPC_, *bla*_VIM_, *bla*_NDM_, *bla*_OXA-48-like_, *bla*_IMP_	n.a.	n.a.	Respiratory specimen	[91]
	ePlex ^®^ Blood Culture Identification Gram Negative Panel (GenMark)	90 min	easy	*bla*_KPC_, *bla*_VIM_, *bla*_NDM_, *bla*_OXA-48-like_, *bla*_OXA-23_, *bla*_IMP_	n.a.	n.a.	blood culture fluid	[92]
	Unyvero (Curetis)	4–5 h	relatively easy	*bla*_KPC_, *bla*_VIM_, *bla*_NDM_, *bla*_OXA-48-like_, *bla*_OXA-23_, *bla*_OXA-24/40_, *bla*_OXA-58_, *bla*_IMP_	n.a.	n.a.	sputum, tissue, bone fragment, pus, blood culture fluid	[93]
	Hyplex SuperBug ID test system (Amplex Diagnostics)	relatively easy		*bla*_KPC_, *bla*_VIM_, *bla*_NDM_, *bla*_OXA-48-like_, *bla*_IMP_	96.7	≥99	sputum, urine, blood culture fluid, swab specimens	[94]
	Luminex xTAG ^®^ assay (Luminex Corporation)	5 h	relatively easy	*bla*_KPC_, *bla*_VIM_, *bla*_NDM_, *bla*_IMI_, *bla*_GES_*, bla*_OXA-23_, *bla*_OXA-51_, *bla*_SEM_, *bla*_VEM_, *bla*_OXA-48-like_, *bla*_IMP_	100	99.4	DNA extracted from bacterial colonies	[95]
	VAPChip ^®^ (EppendorfArray Technologies)	4 h	relatively easy	*bla*_KPC_, *bla*_VIM_, *bla*_OXA-23_, *bla*_OXA-48-like_, *bla*_IMP_, *bla*_OXA-24/40_, *bla*_OXA-58_	100	100	Respiratory sample	[96]
Next generation sequencing	MinION (Oxford Nanopore Technologies)	>8 h	complex	Classes A, B, D	100	100	extracted DNA	[97]

PBC, positive blood culture; sens, sensitivity; spec, specificity; n.a., not available.

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
