# Peer review of "Detection of Multidrug-Resistant Enterobacterales—From ESBLs to Carbapenemases"

_antibiotics, 2021, doi:10.3390/antibiotics10091140_

Round 1

Reviewer 1 Report

Congratulations to the authors for their efforts. The article describes in detail the problem of multidrug resistant enterobacteria, a very important  issue sometimes omitted from the policy of some laboratory/ hospitals / countries.
I suggest the following changes / additions to the authors:
1. list the risk factors (R43) first, then write that the main one is the use of antibiotics
2.the prevalence of colonization can be improved by adding 2-4 references, for instance:

Micle at al.  Recent Advances in Investigation, Prevention, and Management of Healthcare-Associated Infections (HAIs): Resistant Multidrug Strain Colonization and Its Risk Factors in an Intensive Care Unit of a University Hospital

3.Solve:  Error! Reference source not found is written in  R78, 154, 191, R226; my recommendation was to solve them. 
4. Could the authors state (shortly!) therapeutic solutions for some of the MDR strains, for example Acinetobacter baumannii, Klebsiella pneumoniae

Author Response

Comment 1:

 “List the risk factors (R43) first, then write that the main one is the use of antibiotics.”

We changed the order of risk factors as suggested.

Comment 2:

“The prevalence of colonization can be improved by adding 2-4 references, for instance: Micle et al. Recent Advances in Investigation, Prevention, and Management of Healthcare-Associated Infections (HAIs): Resistant Multidrug Strain Colonization and Its Risk Factors in an Intensive Care Unit of a University Hospital”.

We have added further information on the prevalence of colonization with MDR Enterobacterales with a focus on intensive care units. The suggested reference has been added.

Comment 3:

“Solve: Error! Reference source not found is written in R78, 154, 191, R226; my recommendation was to solve them.”

We apologize that the document you received for the reviewing has formatting errors. However, the word document of the review that was offered for download by MDPI was without the mentioned errors on our devices, but when we printed it, we could observe the same mistakes you have mentioned. We therefore suspect technical reasons and replaced cross-references within the text with manual references.

Comment 4:

“Could the authors state (shortly!) therapeutic solutions for some of the MDR strains, for example Acinetobacter baumannii, Klebsiella pneumoniae?”

Thank you very much for your suggestion. The focus of this review are the diagnostic methods for detecting MDR Enterobacterales. Treatment suggestions are beyond the scope of this review, so we prefer not to include any.

Reviewer 2 Report

The authors summarized culture-based molecular techniques for detection of multidrug-resistant Enterobacterales, major cause of hospital acquired infections reviewing preanalytical factors (specimen type, collection site, collection device), screening culture with selective media, susceptibility phenotypic methods, disc diffusion tests, susceptibility genotypic methods.

Major comments

- In general, this manuscript seems to be exhaustive. However, there are so many instances where sentence structure and/or typos detract from the overall readability of the manuscript.

- line 615-616: I guess that a single isolate means bacterial pure cultures. How could NGS analysis of a single isolate offer the tool for culture-free identification of bacteria?

Minor comments

- revise a sentence line 40-42

- revise a sentence line 60

- line 76: group A -> Ambler Class A

- line 79, 154, 191,226: Error! Reference source not found

- add group information by Bush-Jacoby-Medeiros to Figure 1

- Table 1: unify statement for “Target enzyme/gene” using either class name or gene name if possible

- Figure 2A is not mentioned in the manuscript.

- line 232: cefotaxime occurs two times

Author Response

Comment 1

“In general, this manuscript seems to be exhaustive. However, there are so many instances where sentence structure and/or typos detract from the overall readability of the manuscript.”

We apologize that the document you received for review had typos and other issues that affected readability. We have proofread the document again and corrected typos. However, based on comments of the other reviewers, we believe that there are technical issues with the formatting of the document you received. The Word document of the review that was offered for download by MDPI was without the mentioned errors on our devices, but when we printed it, we could observe the same mistakes you have noticed. We therefore replaced cross-references within the text with manual references.

Comment 2:

“Line 615-616: I guess that a single isolate means bacterial pure cultures. How could NGS analysis of a single isolate offer the tool for culture-free identification of bacteria?”

We have corrected the sentence to clarify the statement.

Comment 3:

“revise a sentence line 40-42”

We have corrected the sentence.

Comment 4:

“line 76: group A -> Ambler Class A”

Corrected.

Comment 5:

“line 79, 154, 191,226: Error! Reference source not found”

We suspect some technical issues as stated above (comment 1, reviewer 2) and replaced the cross-references.

Comment 6:

“add group information by Bush-Jacoby-Medeiros to Figure 1”

When adding the Bush-Jacoby-Medeiras classification, figure 1 becomes very complex. For that reason we kept the Ambler classification, which is also used throughout the text.

Comment 7:

“Table 1: unify statement for “Target enzyme/gene” using either class name or gene name if possible.”

Table 1 contains selective culture plates, phenotypic and genotypic tests for detection of beta-lactamases. Therefore, it is difficult to unify (e.g. would not make sense to list all possible carbapenemase gene names for a CPE plate). While molecular methods cannot detect all β-lactamase-encoding genes of the respective Ambler’s class or even within a gene family, cultural methods and others detect enzyme activities, which cannot be differentiated more precisely.

Comment 8:

“Figure 2A is not mentioned in the manuscript.”

We added a cross-reference to Figure 2A.

Comment 9:

“line 232: cefotaxime occurs two times”.

We intended to explain that for the NDP test, the lysed bacteria are incubated in different red phenol solutions. One sample is supplemented with cefotaxime, one with cefotaxime and tazobactam and the third contains no further supplements. We have rephrased the sentence to make the statement clearer.

Reviewer 3 Report

The review article is an excellent review that provides an update on the current methods for detection of Multidrug-resistant Enterobacterales (MDRE) with emphasis on culture-based and molecular techniques, which are valuable for the clinical microbiology laboratory. Overall, the manuscript is clearly written, comprehensive, easy to understand, and is likely to be of interest to a diverse readership, including those interested in antibiotic resistance mechanisms. Although, I do not have any strong criticism for the manuscript, but have few suggestions (see below) as to the content, incorporation of which will further improve the impact of the review on the readers. Apart from this authors need to be congratulated for their commendable effort in putting together this review article.

Minor comment:

  1. At many places in the manuscript, it says reference not found (example Line number- 78, 191, and so on). Not sure whether references are missing there or due to some other technical error.
  2. For figure 4, figure legend should be more descriptive providing more details.
  3. The spacing problems between the paragraphs better proofreading of text is required before acceptance.

Author Response

“At many places in the manuscript, it says reference not found (example Line number- 78, 191, and so on). Not sure whether references are missing there or due to some other technical error.”

We apologize that the document you received for the reviewing has formatting errors. However, the Word document of the review that was offered for download by MDPI was without the mentioned errors on our devices, but when we printed it, we could observe the same mistakes you have mentioned. We therefore suspect technical reasons and replaced cross-references within the text with manual references.

Comment 2:

“For figure 4, figure legend should be more descriptive providing more details.”

We added further details to the legend of figure 4.

 Comment 3:

“The spacing problems between the paragraphs better proofreading of text is required before acceptance.”

We apologize that the document you received for review had typos and other issues that affected readability. We have proofread the document again and corrected typos.